# Influence of Dispersion and Orientation on Polyamide-6 Cellulose Nanocomposites Manufactured through Liquid-Assisted Extrusion

**DOI:** 10.3390/nano12050818

**Published:** 2022-02-28

**Authors:** Luísa Rosenstock Völtz, Shiyu Geng, Anita Teleman, Kristiina Oksman

**Affiliations:** 1Division of Materials Science, Department of Engineering Sciences and Mathematics, Luleå University of Technology, SE-97187 Luleå, Sweden; luisa.voltz@ltu.se (L.R.V.); shiyu.geng@ltu.se (S.G.); 2Wallenberg Wood Science Center (WWSC), Luleå University of Technology, SE-97187 Luleå, Sweden; 3RISE Research Institutes of Sweden, SE-11486 Stockholm, Sweden; anita.teleman@ri.se; 4Department of Mechanical & Industrial Engineering (MIE), University of Toronto, Toronto, ON M5S 3G8, Canada

**Keywords:** nanocellulose, polyamide 6, nanocomposites, liquid-assisted extrusion, anisotropy

## Abstract

In this study, the possibility of adding nanocellulose and its dispersion to polyamide 6 (PA6), a polymer with a high melting temperature, is investigated using melt extrusion. The main challenges of the extrusion of these materials are achieving a homogeneous dispersion and avoiding the thermal degradation of nanocellulose. These challenges are overcome by using an aqueous suspension of never-dried nanocellulose, which is pumped into the molten polymer without any chemical modification or drying. Furthermore, polyethylene glycol is tested as a dispersant for nanocellulose. The dispersion, thermal degradation, and mechanical and viscoelastic properties of the nanocomposites are studied. The results show that the dispersant has a positive impact on the dispersion of nanocellulose and that the liquid-assisted melt compounding does not cause the degradation of nanocellulose. The addition of only 0.5 wt.% nanocellulose increases the stiffness of the neat polyamide 6 from 2 to 2.3 GPa and shifts the tan δ peak toward higher temperatures, indicating an interaction between PA6 and nanocellulose. The addition of the dispersant decreases the strength and modulus but has a significant effect on the elongation and toughness. To further enhance the mechanical properties of the nanocomposites, solid-state drawing is used to create an oriented structure in the polymer and nanocomposites. The orientation greatly improves its mechanical properties, and the oriented nanocomposite with polyethylene glycol as dispersant exhibits the best alignment and properties: with orientation, the strength increases from 52 to 221 MPa, modulus from 1.4 to 2.8 GPa, and toughness 30 to 33 MJ m^−3^ in a draw ratio of 2.5. This study shows that nanocellulose can be added to PA6 by liquid-assisted extrusion with good dispersion and without degradation and that the orientation of the structure is a highly-effective method for producing thermoplastic nanocomposites with excellent mechanical properties.

## 1. Introduction

Polyamide 6 (PA6), also called nylon 6, is a thermoplastic polymer used in a variety of applications owing to its excellent mechanical and thermal properties, as well as good adhesion and attractive surface appearance [1]. It has been reported that PA6 can be successfully reinforced with glass fibers [2], clay [3,4,5], and cellulose [6,7] to improve its properties. In the 1990s, studies on PA6-nanocomposites were carried out by Toyota [4,5] where PA6 was polymerized using montmorillonite (MMT) as a catalyst. The addition of MMT (4.7 wt.%) resulted in an increase in the tensile strength of these composites from 69 to 97 MPa and in their tensile modulus from 1.1 to 1.9 GPa. The use of cellulose nanocrystals (CNCs) [8,9,10,11,12] and cellulose nanofibers (CNFs) [8,13,14,15] as reinforcements for polymer composites have also been studied in PA6.

The main challenge in their processing is associated with an effective dispersion and distribution of the cellulose nanomaterials in the polymer matrix, owing to their tendency to agglomerate and their thermal instability at temperatures above 200 °C [16]. To overcome these challenges, high-energy processes [11,13], and chemical modifications [12,15,16] have been used. For example, ball-milling and cryo-milling [11] were used to produce the polymer and cellulose in powder form to facilitate the mixing and dispersion of cellulose with PA6. However, these processes are time- and energy-intensive, rendering the large-scale production difficult. Regarding chemical modification [15], cellulose has been modified to enhance its thermal stability and dispersion in PA6. Acetylated cellulose (10 wt.%) was fibrillated during the extrusion process, resulting in improved mechanical properties; however, a chemical modification requires solvent exchange and the use of harsh chemicals. Solvent casting [12,16] is another process used for the preparation of nanocomposites; through it, a good dispersion can be achieved, but both the polymer and nanocellulose (NC) must be dissolved and dispersed in the same solvent, followed by a casting and evaporation of the solvents used. This type of process is useful on a laboratory scale but is not environmentally friendly or scalable. Most of the reported studies have used dried NC in PA6 [8,11,13,14,15,16,17], at rather high concentrations [13,14]. However, even if NC is well distributed in the polymer phase, it is not dispersed at the nanoscale.

A successful approach to incorporate the nanomaterial into the polymer is liquid-assisted extrusion, in which the nanomaterial is first dispersed in a water suspension that is pumped into the extruder without any chemical modifications, as reported previously by our research group [18,19,20], with polylactic acid (PLA). Clemons and co-workers also tested this process with PA6 [9]. These studies demonstrated that this process leads to well-dispersed NC; however, fast water-evaporation can also lead to agglomeration. Peng et al. [9] prepared PA6/CNC nanocomposites (with CNC concentrations ranging from 0.5 to 3.5 wt.%) using this process and showed a good dispersion of the CNCs; however, the mechanical properties were not improved, and the CNCs behaved as more of nucleation agents for crystallization and foaming than of reinforcement.

In addition to the use of nanomaterials to improve the mechanical properties, the alignment of the polymer molecular chains and reinforcing phase is an effective way to enhance the properties of nanocomposites [21,22]. Previous studies on PLA-cellulose nanocomposites have shown that solid-state drawing (SSD) results in an enhancement in stiffness, strength, and toughness [21,23]. There are also some studies available on the drawing of polyamides [24,25,26]; however, to the best of our knowledge, the orientation of PA6–cellulose nanocomposites have not been previously reported.

This work focused on PA6-nanocomposite processing using liquid-assisted extrusion. The effects of processing and the use of a dispersant, i.e., polyethylene glycol, were investigated in terms of the dispersion of NC and its effect on the crystallinity and thermal and mechanical properties. Furthermore, we studied the effect of SSD, used to align the PA6 and PA6-nanocomposites, on the above-mentioned properties and structure. 

## 2. Materials and Methods

### 2.1. Materials 

PA6 SCANAMID 6 B12, purchased from Polykemi AB (Ystad, Sweden), was dried at 60 °C overnight, prior to its use. The melt-flow index and density of the polymer were 3 g/10 min (235 °C/2.16 kg) and 1130 kg/m^3^, respectively, as reported by the supplier. A commercial micro-fibrillated cellulose (MFC Grade 1) with 19 wt.% concentration was kindly supplied by Stora Enso Oyj (Helsinki, Finland); this material consists of nanosized cellulose fibers in a water suspension and is hereinafter referred to as NC. Polyethylene glycol (PEG, Mw 600 g/mol) was purchased from VWR International AB (Karlskoga, Sweden) and was used as a dispersant for NC.

### 2.2. Methods

#### 2.2.1. Preparation of NC Suspensions

The NC suspensions for the extrusion process were prepared by mixing commercial NC (2 wt.%) in distilled water and 0 or 10 wt.% PEG, with the aid of a magnetic stirrer. Afterward, the NC suspensions (2 wt.%) are further dispersed using an Ultra-Turrax (IKA T25, IKA Werke GmbH & Co. KG, Oxfordshire, UK) for 10 min, as shown in Appendix A. The width of cellulose fibrils in the NC was measured using tapping-mode atomic force microscopy (Veeco MultiMode scanning probe, Santa Barbara, CA, USA). The sample was prepared by depositing a drop of diluted NC suspension (0.01 wt.%) on a fresh mica plate and letting it dry in a vacuum oven at 80 °C for 5 min. The measurements were recorded from images at height, using Nanoscope V software, and the average width values presented (Appendix A), varying between 5 and 40 nm, were based on 40 different measurements. The viscosities of the suspensions with different NC concentrations were measured using a Vibro viscometer (SV-10, A&D Company Limited, Tokyo, Japan). This method is used as a comparative metric of the viscosity with different NC contents (1.5, 2, and 3 wt.%), as shown in Appendix A. The suspension viscosity increases with the increase in cellulose concentration, where 1.5 wt.% cellulose content results in a viscosity of approximately 400 mPa·s, and 3.0 wt.% cellulose in the suspension increases the viscosity almost 10-fold, to 3500 mPa·s. Subsequently, the NC suspensions with different concentrations were passed through the pump, and the flow rate was measured. More accurate measurements were obtained for 1.5 and 2 wt.% NC suspensions. Optical microscopy (OM) (Eclipse LV100POL, Nikon, Kanagawa, Japan) was used to observe the morphology of the fibers (Appendix A) by placing a drop of the NC suspension (0.01 wt.%) on a glass slide.

#### 2.2.2. Liquid-Assisted Extrusion and Film Preparation 

Suspensions containing high-concentrations of NC are difficult to pump because NC suspensions have high viscosity (as explained in Section 2.2.1) even at low-concentrations, owing to the fiber length. Suspension viscosity is a limiting factor for the NC content in both the suspension and final composite. The water content should be as low as possible to guarantee slow and complete evaporation during extrusion. An overly fast water-evaporation can lead to agglomeration; moreover, the water remaining in the nanocomposite tends to interact with the polymer chains. Thus, an equilibrium must be found between the viscosity and liquid content. In this study, the maximum NC content was 2 wt.%, which was limited by the viscosity. To guarantee successful removal of water, a co-rotating twin-screw extruder W&P ZSK-18 MEGALab (Coperion, Stuttgart, Germany) is used because of its excellent degassing system, as shown in Figure 1a. A high-pressure syringe pump 500 D (Teledyne Isco, Lincoln, NE, USA) was used to feed the NC-containing suspension into the melt polymer (Figure 1a). The feeding rate of PA6 was kept constant at 1.2 kg/h, and the screw speed was 200 rpm for all compounding trials. The used temperature setup (starting from 232 °C in the processing section to melt the PA6 and reduced to 225 °C) and used screw design in the extrusion process are shown in Figure 1a. The neat PA6 used as a reference material was prepared similarly to the PA6-nanocomposites rather than the NC suspension—only water was used as the pumping medium. The vaporized water was removed via atmospheric and vacuum venting. Table 1 lists the nanocomposites, their coding, compositions, as well as the feed and water evaporation rates.

A rectangular die was used, and the PA6 and PA6-nanocomposites were collected and dried in an oven (40 °C for 12 h). Subsequently, approximately 4–5 g of the material was placed between Teflon-covered steel plates, pre-heated for 120 s at 225 °C and compression-molded at a pressure of 2.2 MPa for 60 s using a LabEcon 300 Fontijne Grotness press (Vlaardigen, The Netherlands). The films were then cooled to room-temperature (25 °C) under pressure (approximately 5 min). The compression molding step was repeated to ensure homogeneous film composition as it was observed that the films were not completely homogeneous, and it was believed that the reason was the low molecular weight PEG. Therefore, the large films were cut into four smaller pieces which were melted again at 225 °C for 60 s and compression molded for another 60 s at the same pressure and temperature as the previous films. The second compression molding resulted in better film quality.

#### 2.2.3. SSD

SSD of the films was performed using a universal tensile testing machine (Shimadzu Autograph AGX, Kyoto, Japan) equipped with a temperature chamber and a load cell of 1 kN (Figure 1a). The drawing was performed at 140 °C, a rate of 100 mm/min, and the maximum draw ratio (DR) of 2.5 (the final gauge length divided by the original gauge length). The drawing temperature is chosen according to the DSC analysis of neat PA6 shown in Figure 1b, above the glass transition temperature (≈65 °C) [27] and below the cold crystallization temperature. The PA6-nanocomposite samples before and after drawing are shown in Figure 1c. The samples after SSD are marked with “O” before the sample name. 

### 2.3. Characterizations

Optical and polarized optical microscopes (OM and POM, respectively) (Eclipse LV100POL, Nikon, Kanagawa, Japan) were used to investigate the dispersion of NC in the suspension, nanocomposite films, as well as the birefringence caused by the orientation of the PA6 and PA6-nanocomposite films. POM equipped with a Linkam hot-stage THM600 (Tadworth, UK) to study the crystallization behavior of PA6 and PA6-nanocomposites films. The films were heated between glass covers to 232 °C and then cooled to 200 °C which is a crystallization temperature of PA6. The nucleation and crystal growth were studied between a time interval of 0 to 120 s. High-resolution scanning electron microscopy (SEM) (FEI Magellan 4000 XHR-SEM, Thermo Fisher Scientific, Hillsboro, OR, USA) was used to analyze the effects of the SSD on the surface of the PA6 and PA6-nanocomposite films, at an acceleration voltage of 3 kV. The film was placed on the holder, and the surface was sputter-coated with platinum using a Leica EM ACE200 sputtering instrument (Wetzlar, Germany, 15 nm thick layer), to avoid charging. 

The thermal stability and onset degradation temperature of the PA6 and PA6-nanocomposites were measured using thermogravimetric analysis (TGA) (Q500, TA Instruments, New Castle, DE, USA) under a nitrogen flow, at a heating rate of 10 °C/min within a temperature range of 0–600 °C. Differential scanning calorimetry (DSC) (Mettler Toledo DSC821, Columbus, OH, USA) was used to study crystallinity and melt temperatures of the materials before and after SSD. The sample, weighing around 4–6 mg, was placed in a hermetic aluminum pan and tested under a nitrogen atmosphere from 25 to 260 °C, at a heating rate of 10 °C/min. The degree of crystallinity (X_c_) was calculated using the enthalpy of 100% crystalline PA6 (230 J/g). 

The mechanical properties of the materials (before and after SSD) were tested under controlled temperature and low-humidity (25 °C and 15% RH) using a Shimadzu Autograph AGX universal testing machine (Kyoto, Japan) with a 1 kN load cell. The samples were oven-dried prior to testing. The tests were performed with a gauge length of 20 mm and a crosshead speed of 5 mm/min. At least five specimens were tested for each sample. For the tested samples, a statistical analysis at a 5% significance level, based on the ANOVA and Tukey-HSD multiple comparison tests, was used to analyze the results. 

The storage modulus and tan δ peak as a function of the temperature of the PA6 and PA6-nanocomposites (before and after SSD) were determined using DMA Q800 (TA Instruments New Castle, DE, USA) using a temperature ramp and single frequency of 1 Hz. The test was performed in tensile mode, within a temperature range between –50 and 150 °C, at a heating rate of 3 °C/min; the materials were oven-dried before the test. 

FT-IR spectroscopy was used to evaluate the effect of SSD on the PA6-nanocomposites. The films were placed in a VERTEX 80 (Bruker, Billerica, MA, USA) FT-IR spectrophotometer with a range of 500–3500 cm^−1^, and a 256-scan resolution was used. The curves were normalized by the highest intensity to minimize errors owing to differences in thickness. 

The crystal type and crystal size of the PA6 and PA6-nanocomposites (before and after SSD) were studied using an X-ray diffractometer (XRD) (PANalytical Empyrean X-ray diffractometer, Malvern Instruments, Malvern, UK) with CuKα radiation (a wavelength of 1.540 Å), generated at an acceleration voltage of 45 kV and current of 40 mA. The 2θ scan angle range was 5°–40°. The apparent crystallite size was obtained from the half-width of the 2θ peaks using Scherrer’s Equation (1) [28]:(1)D=kλ/β cosθ 
where *D* is the crystal size, k is a constant that is 0.89 [28], *λ* is the wavelength of the X-ray, *β* is the width of the peak at half the maximum intensity, and *θ* is the Bragg’s angle. The degree of orientation of the PA6 and PA6-nanocomposites was studied using wide-angle X-ray scattering (WAXS) analysis on an Anton Paar SAXSpoint 2.0 system (Anton Paar, Graz, Austria) equipped with a Microsource X-ray source (Cu K-alpha radiation, a wavelength of 0.15418 nm) and a Dectris 2D CMOS Eiger R 1M detector with a pixel size of 75 × 75 µm^2^. All measurements were performed with a beam size of approximately 500 µm in diameter and a beam path pressure of approximately 1–2 mbar. All samples were mounted on a Sampler for Solids 10 × 10 mm^2^ (Anton Paar, Graz, Austria) holder. The transmittance was determined and used for scaling the intensities. The software used for instrument control was SAXS drive version 2.01.224 (Anton Paar, Graz, Austria), and post-acquisition data-processing was performed using SAXS analysis version 4.01.047 (Anton Paar, Graz, Austria). The orientation index (*f_c_*) of the PA6 and PA6-nanocomposites was calculated based on the intensity distributions of the azimuthal angle using the following equation:(2)$$f_{c}=\left(180{}^{\circ}-FWHM\right)/180{}^{\circ},$$
where *FWHM* is the full width at half maximum of the azimuthal angle distribution.

## 3. Results and Discussions

### 3.1. Effect of the Addition of NC and NC/PEG to PA6 

The dispersion of NC in PA6 was studied using OM (Figure 2). The presence of dark spots (indicated by the circles in Figure 2a) is observed, which can be indicative of NC agglomeration, and the number of these spots decreases with the addition of PEG (5 wt.%), as illustrated in Figure 2b. PEG is expected to protect NC during the water-evaporation step, reducing agglomeration.

The thermal properties of PA6 and its nanocomposites are shown in Appendix A. There was no difference in the degradation temperature between PA6 and PA6/0.5NC, which suggests that NC (degradation onset around 273 °C) can be used with polymers with a high melting temperature, such as PA6. The liquid suspension decreases the processing temperature [9], resulting in lower processing temperatures in the last zones (shown in Figure 1a), which is beneficial for NC.

The degree of crystallinity of PA6/0.5NC (28%) is higher than that of PA6 (21%), as listed in Appendix A. The addition of PEG/NC had a small effect on crystallinity (25%), explained by the similar polarities of PEG and PA6, which decreased the degree of crystallinity [29]. The PA6 shows two melting endotherms, *T_m1_* at ~205 °C and *T_m2_* at 218 °C, as illustrated in Appendix A. Khanna [27] found that the two melting-temperature peaks at 215 °C and 225 °C in nylon 6 are associated with two different crystal types, i.e., γ and α crystals, respectively. The double melting peaks observed in the DSC results are likely associated with these two crystal forms.

The addition of only 0.5 wt.% NC increases the *E* modulus from 2.0 to 2.3 GPa, while the addition of PEG/NC reduces it to 1.4 GPa, as shown in Appendix A. In terms of strength, no significant differences were observed for PA6 and PA6/0.5NC; however, the use of PEG reduced the strength of the nanocomposite. Similarly, in the work performed by Peng et al. [8], the addition of 0.5 wt.% CNCs in PA6 did not enhance the mechanical properties. The addition of PEG contributes to an increase in the elongation at break, from 19 to 63%, resulting in an increase in toughness, from 10 to 29 MJ m^−3^, as shown in Appendix A. Oksman et al. [20] reported that the addition of PEG (15 wt.%) as an additive in cellulose nanowhisker suspension improved the dispersion of NC in PLA with an increase in elongation at break but with a decrease in modulus and strength when compared with the corresponding nanocomposite. 

The viscoelastic behaviors of PA6 and PA6-nanocomposites are shown in Figure 3. The α-relaxation peak assigned as the tan δ peak (≈65 °C) is related to the breaking of hydrogen bonds between the polymer chains, inducing chain motion in the amorphous region [30]. The storage modulus (*E′*) of PA6 and its nanocomposites remain at a similar level, while the tan δ peak moves to a higher temperature (from 61 to 69 °C) with the addition of NC, indicating an interaction between PA6 and NC. The addition of PEG/NC did not affect the tan δ peak position, in contrast to the addition of NC only. The use of PEG was expected to change the tan δ peak to a lower temperature due to the plasticizing effect of PEG [31], but the NC in this material counteracts the effect and therefore the peak position is not affected compared to PA6.

The 1D-XRD scattering patterns, shown in Appendix A, reveal that the two main characteristic diffraction peaks for the PA6 and PA6–nanocomposite are located around 20.2° and 23.7°. According to the literature [32], the α-phase peaks for PA6 at 20.1° (200) and 23.8° (002/202) are related to the α_1_ and α_2_ peaks, respectively, and 21.3° (200/001) is related to the γ-phase. Other researchers [33] showed that α-phase and γ-phase can coexist in PA6 and that the γ-phase presents peaks at 22° (001) and 23° (200/201). The α-phase has a monoclinic structure, where polyamide molecules form an antiparallel chain arrangement hydrogen bonding. The γ-phase is also monoclinic, but the polyamide molecules adopt parallel chain arrangement hydrogen bonding. In Appendix A, no peaks from NC and PEG are visible owing to their low concentrations in the PA6-nanocomposites. However, the addition of NC increased the apparent crystallite size in PA6, from 5.32 to 7.79 nm, and the addition of PEG in the nanocomposites further increased the size to 6.70 nm. 

In addition, the crystallization of the PA6 and its nanocomposites was studied with POM and the results are shown in Appendix A. The images show fast crystallization behavior of the neat PA6, as well as the nanocomposites. The crystallization temperature for PA6 is at 200 °C and the visible nucleation (in the used magnification) of the crystallites is starting already after 40 s and appears to be completed at around 90 s, after that no difference is seen. The crystal size is very small, and no large crystals are visible in any of the materials. In addition, no difference of the nucleation and crystallization rate among the materials were visible, indicating that the NC is not acting as a nucleating agent. The exact crystal size cannot be defined from the POM images because of the small size. 

### 3.2. Effect of Orientation on PA6 and PA6-Nanocomposites

Figure 4 shows cross-polarized optical micrographs, with the orientation of the films at 0° and 45° to the cross-polarizers. The micrographs confirm the orientation of the films as their birefringence initially increases and reaches maximum power at 0° (Figure 4a) and subsequently decreases to a minimum at 45° (Figure 4b) as the film orientation is positioned parallel to one of the polarizers. All the materials show non-homogeneous birefringence, expressed by the heterogeneous orientation of the polymer chains, as indicated by the different colors.

As shown in Figure 4a, the OPA6 and OPA6-nanocomposites have defects. OPA6 presents the highest birefringence with a small number of defects, followed by OPA6/5PEG/0.5NC. Most defects are visible in PA6/0.5NC. Because OPA6 also presents a small number of defects, their presence could be associated with the excessive stretching during SSD, and the large defects seen with the addition of NC could be related to NC agglomeration. This further confirms the improvement in dispersion with the addition of PEG. Barkoula et al. [34] discussed that drawing polar polymers, such as PA6, below their melting temperature is difficult because of the strong hydrogen bonds between the chains and their folding, creating a resistance of the crystallite to deformation and limiting the drawing ratio. To clarify the defects shown in the POM, the OPA6 and OPA6-nanocomposite film surfaces are examined using SEM, as shown in Figure 5. The micrographs confirm the presence of cavities on the surface of the OPA6 and OPA6-nanocomposite films. More accentuated cavities are observed in the OPA6/0.5NC films than in the nanocomposite with PEG. Corroborating the POM images, the lower number of cavities in OPA6/5PEG/0.5NC is explained by the better distribution of NC in PA6. The cavities in OPA6 and its nanocomposites can also be attributed to the excessive stretching of the film. 

The DSC thermographs in Figure 6 show that the γ phase (*T_m_*_1_) is more pronounced after the SSD for all samples. Furthermore, the degree of crystallinity increased, and both melting-peaks shifted to higher temperatures with the orientation; a relatively high increment in the intensity was observed for *T_m_*_1_ (Appendix A). The increase in crystallinity after the orientation process is explained by both the orientation of molecules and their much closer packing, allowing for the crystallites to be formed from the previously amorphous region [35].

In Figure 7, the mechanical properties of PA6 and PA6-nanocomposites before and after SSD are presented. The SSD has a significant effect on the strength, modulus, and toughness of OPA6 and OPA6-nanocomposites, when compared with non-oriented samples. For OPA6/0.5NC, the modulus improved by 30% (from 2.3 to 3.0 GPa), strength increased by 258% (from 60 to 215 MPa), and toughness improved by 140% (10 to 24 MJ/m^3^), when compared with the non-oriented PA6/0.5NC. A comparison of OPA6/5PEG/0.5NC and PA6/5PEG/0.5NC shows that after SSD, strength improved by 325% (from 52 to 221 MPa) and the modulus doubled (from 1.4 to 2.8 GPa). 

A comparison of the OPA6 and OPA6-nanocomposites showed that the addition of PEG (OPA6/5PEG/0.5NC) resulted in the highest tensile strength (221 MPa) and toughness, as well as comparable modulus and elongation at break (Appendix A). The increase in the strength of OPA6/0.5NC and OPA6/5PEG/0.5NC, compared with OPA6, demonstrates that in addition to the presence of defects, the alignment of NC contributes to the enhancement in the mechanical properties. Even though SSD tends to increase the elongation at break, the opposite behavior observed here can be attributed to the increase in the degree of orientation and fraction of the rigid amorphous phase [36], supported by the increase in crystallinity observed in the DSC results. Bugatti et al. [22] have also reported a decrease in elongation at break after drawing poly-(ε-caprolactone) nanocomposites. Given that, the high tensile strength attained through the orientation of the polymer chains and nanoparticles is very favorable to produce tough and stiff materials.

The dynamic mechanical behavior is illustrated in Figure 8, which shows that the OPA6 and OPA6-nanocomposites present higher storage modulus values than the non-oriented PA6 and PA6-nanocomposites, as expected according to the tensile properties. In addition, the tan δ peak shifts to high temperatures after the SSD (from 60 to 90 °C). The shift of the tan δ peak to a high temperature can be attributed to the strain-induced crystallization and constrained-oriented amorphous phase, as noted by Mai et al. [21] after the SSD of PLA, as well as to the increment in the degree of crystallinity [32], supported by the DSC results. Similarly, the amplitude of the tan δ peak increases after SSD, indicating an enhancement in the material damping and that oriented PA6 and PA6-nanocomposites can absorb and dissipate the energy better, resulting in an increased toughness, as observed from the mechanical properties. Penel-Pierron et al. [24] suggested that the difference in amplitude is caused by the different arrangements of the crystallites between different samples, modifying their damping behavior. 

Appendix A, shows the FT-IR spectra of PA6-nanocomposites and OPA6-nanocomposites. It is possible to see that both types of crystals (α and γ) are presented before and after SSD, the peak around 1121 cm^−1^ is associated with γ crystals, and a higher-intensity peak is found after the SSD. A slight shift from 1238 to 1236 after SSD can be attributed to the vibration of amide III group in α-crystals (peak at 1238) to γ-crystals (peak in 1232), also a peak in 1530 cm^−1^ related to amide II group in γ-crystals is found for oriented samples [37]. In addition, the non-oriented PA6-nanocomposites show higher intensity peaks at ~3300 cm^−1^ than oriented PA6-nanocomposites. This peak is related to the hydrogen-bonded N–H stretch in the crystalline phase [38]. During drawing, the hydrogen-bond network is broken and formed constantly in a dynamic system; therefore, it can be suggested that the hydrogen bonds are not fully formed during the drawing of PA6-nanocomposites, as reported previously by Xu et al. [32] during the stretching of the β-PA6 film. 

Figure 9 illustrates the 1D-XRD results for the non-oriented and oriented PA6 and PA6-nanocomposites. The oriented PA6 and PA6-nanocomposites present broader peaks than the non-oriented ones. Moreover, for the OPA6-nanocomposites, the peak at 2*θ* = 20.3° almost disappeared and the peak at 2*θ* = 23.7° shifted toward low Bragg’s angles (≈23.2°, Appendix A). This could be indicative of the crystal-to-crystal transition from the α-phase (2*θ* = 23.7°) to the γ-phase (2*θ* = 23°) [27], corroborating the results of DSC. In fact, the apparent crystallite size of all oriented nanocomposites decreased, as suggested by the peak broadening. Shi and co-workers [26] attributed the decrease in the crystallite size after the hot drawing of PA6 to the slipping and rupture of the lamellae in the spherulite.

To confirm the effect of SSD found using the results of 1D-XRD, a 2D-WAXS analysis was performed (Appendix A). The crystal peaks of PA6 were observed in all the samples (Appendix A). No peaks of NC and no distinct peaks of PEG were observed, which is consistent with the 1D-XRD results. A mechanically induced phase transition of a part of PA6 from the α-phase to the γ-phase can be suggested to occur in the oriented nanocomposites because the shift of the peak at 23.7° to a lower angle, corroborating the results of 1D-XRD. The PA6 orientation was estimated from the orientation index (*f_c_*) calculated at two peaks: 2*θ* = 20.1° and 23.7° (Appendix A). The *f_c_* of PA6 is higher by 0.53–0.6 than those of PA6/0.5NC and PA6/5PEG/0.5NC. It is difficult to explain the high-orientation degree of PA6 after compression molding, but it may be suggested that crystal formation during cooling is stacking in a preferential orientation, while the presence of NC hinders this preferential orientation. However, the 2D-XRD analysis shows that the orientation index for the OPA6-nanocomposites is higher than that for OPA6, in the range of 0.84–0.86. Although the results do not prove the orientation of the NC because the peaks for the NC cannot be observed and the diffraction peaks are only related to PA6, the enhanced mechanical properties of OPA6/0.5NC and OPA6/5PEG/0.5NC can be an indicator of a slight orientation of NC in the OPA6-nanocomposites. In addition, different intensities of the diffraction peaks observed in 1D-XRD and 2D-XRD can be attributed to different crystal orientations along the film.

## 4. Conclusions

In this work, we showed that liquid-assisted extrusion and the use of a PEG dispersant resulted in good dispersion and that the high-temperature process (232 °C) did not degrade the NC. The addition of a small amount of NC had a positive effect on the modulus, and the interaction between the NC and PA6 was excellent, as confirmed by DMA. SSD had a significant effect on the strength, stiffness, and toughness of PA6 and its nanocomposites. OPA6/5PEG/0.5NC was the nanocomposite with the best properties: a strength of 221 MPa, modulus of 2.8 GPa, and elongation at break of 23%, given the toughness of approximately 33 MJ m^−3^ with an orientation factor of 0.86. As the addition of 5 wt.% PEG improved the dispersion of the NC, increased the elongation and toughness but decreased the strength and modulus of non-oriented PA6-nanocomposites, a future study on the optimum concentration should be performed. In our findings, the use of never-dried nanocellulose in PA6 and the alignment of nanocomposites contributed to the improvement of mechanical properties.

## Figures and Tables

**Figure 1 nanomaterials-12-00818-f001:**
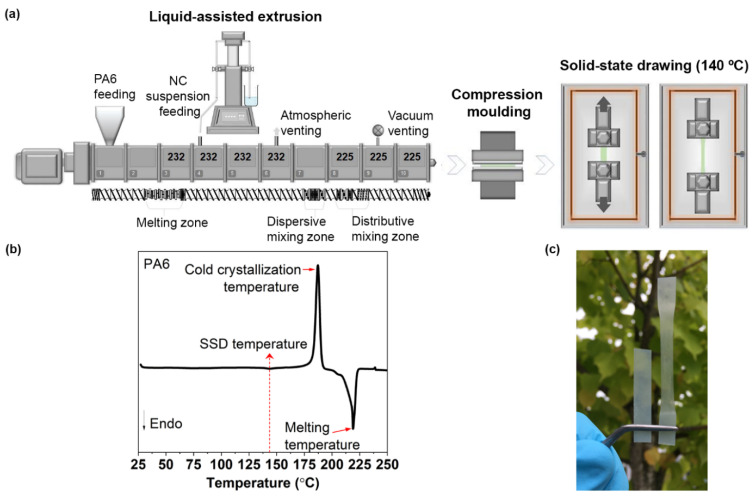
(**a**) Schematic of the manufacturing process for PA6 and PA6-nanocomposites: liquid-assisted extrusion using a co-rotating twin-screw extruder (with the profile temperature in °C), compression molding, and SSD at 140 °C. (**b**) DSC thermogram of PA6 indicating the drawing temperature. (**c**) Image of the nanocomposite film before (left) and after (right) SSD.

**Figure 2 nanomaterials-12-00818-f002:**
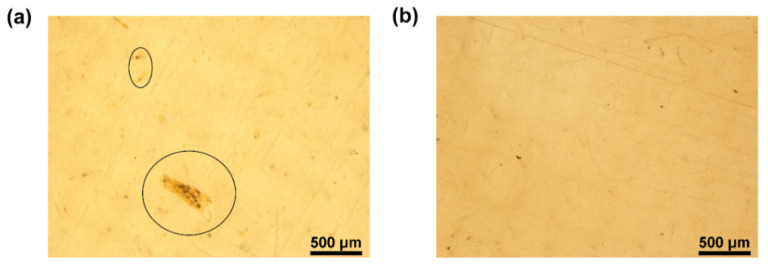
OM images of non-oriented PA6-nanocomposites films (**a**) PA6/0.5NC and (**b**) PA6/5PEG/0.5NC.

**Figure 3 nanomaterials-12-00818-f003:**
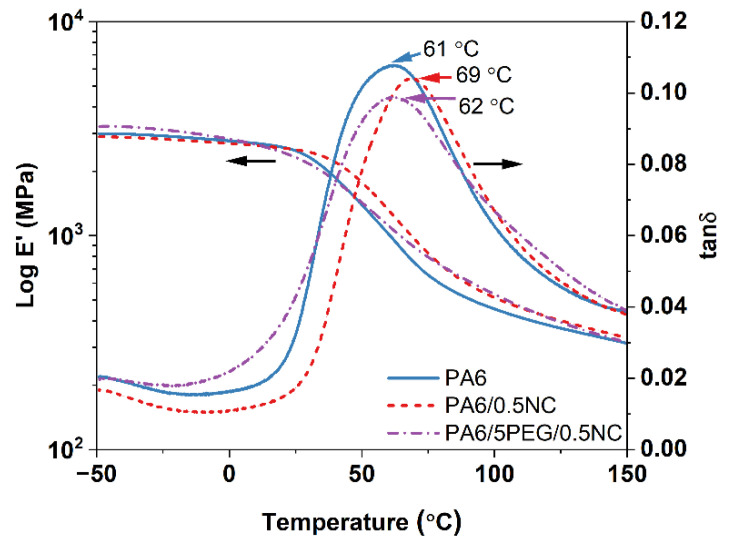
Storage modulus and tan δ as a function of temperature for PA6 and PA6-nanocomposites.

**Figure 4 nanomaterials-12-00818-f004:**
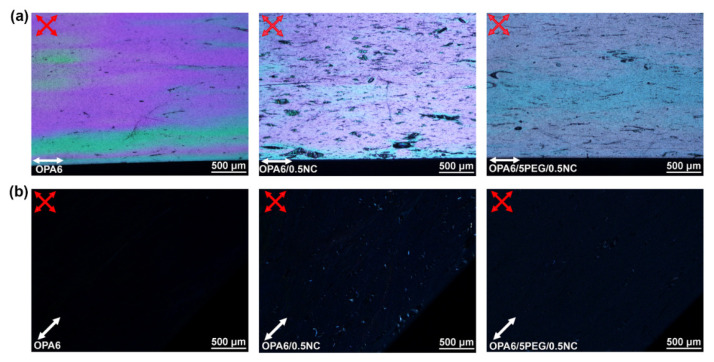
Cross-polarized optical microscopy images, sample positioned in (**a**) 0° and (**b**) 45°. The white arrows indicate the angle-direction of the film position and orientation, and the red arrows indicate the angle of the polarizers.

**Figure 5 nanomaterials-12-00818-f005:**
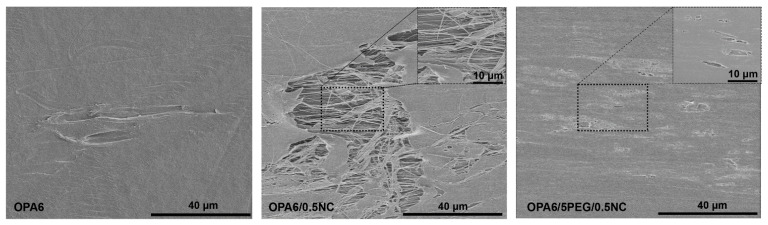
SEM micrographs of OPA6 and OPA6-nanocomposite surfaces to analyze the defects after SSD.

**Figure 6 nanomaterials-12-00818-f006:**
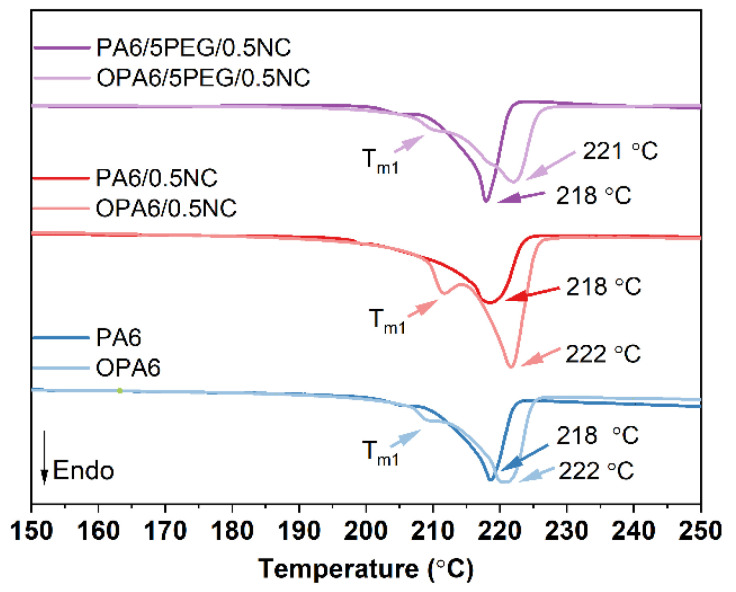
DSC thermograms of the PA6 and PA6-nanocomposites before and after SSD.

**Figure 7 nanomaterials-12-00818-f007:**
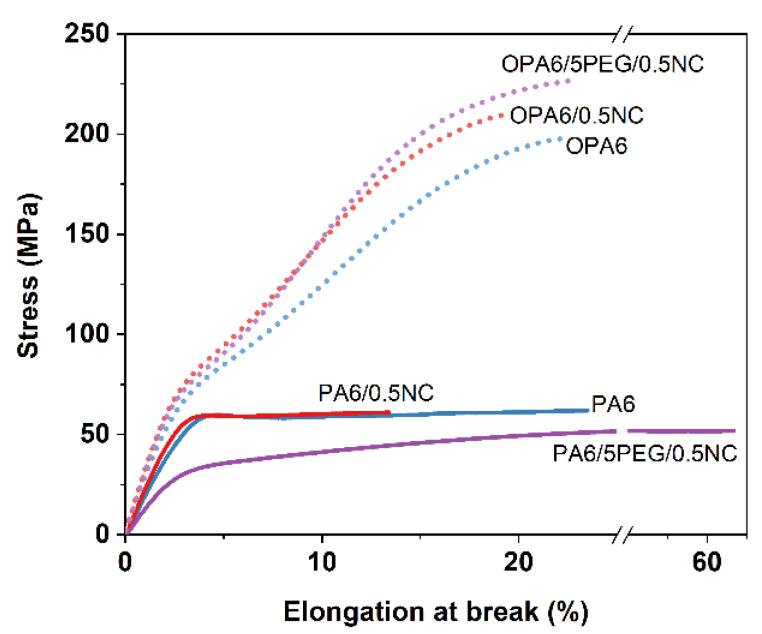
Representative tensile testing curves of non-orientated and orientated PA6 and PA6-nanocomposites.

**Figure 8 nanomaterials-12-00818-f008:**
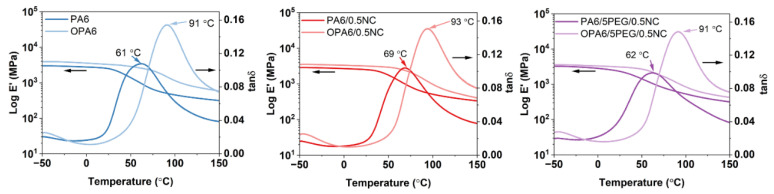
Storage modulus and tan δ as a function of temperature for the non-oriented and oriented PA6 and PA6-nanocomposites.

**Figure 9 nanomaterials-12-00818-f009:**
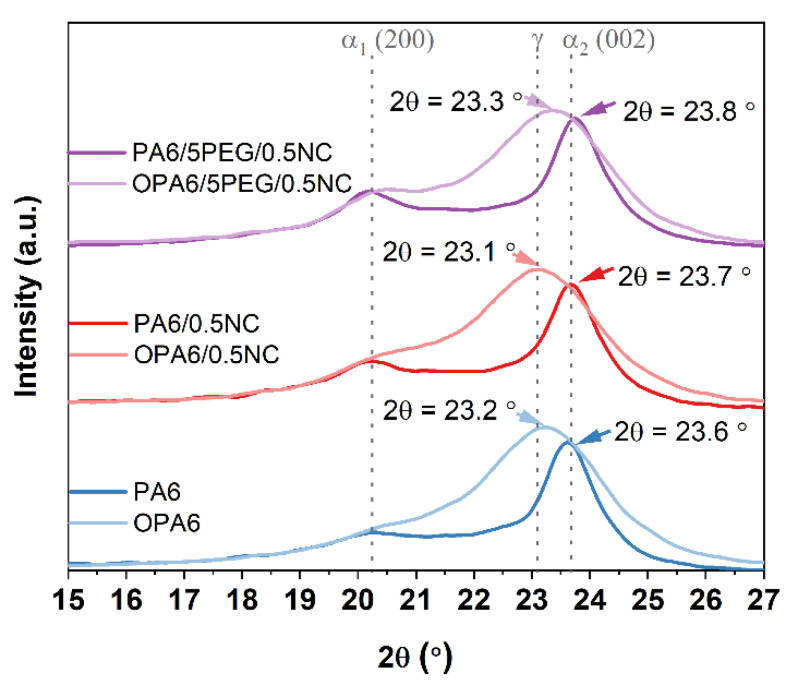
The 1D-XRD scattering patterns of the PA6 and its nanocomposites before and after orientation.

**Table 1 nanomaterials-12-00818-t001:** Sample coding for the nanocomposites, NC and PEG concentrations, pumping rate of the water suspension with NC and PEG, and evaporation rate of the water phase.

Sample Code	NC (wt.%)	PEG (wt.%)	Pumping Rate (mL/min)	Evaporation (mL/min)
PA6	0	0	5.0	5.0
PA6/0.5NC	0.5	0	5.0	4.9
PA6/5PEG/0.5NC	0.5	5	5.3	4.6

## Data Availability

Data are available on request.

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
