# Peer review of "Influence of Dispersion and Orientation on Polyamide-6 Cellulose Nanocomposites Manufactured through Liquid-Assisted Extrusion"

_nanomaterials, 2022, doi:10.3390/nano12050818_

Round 1

Reviewer 1 Report

The authors presented the results of novel combination of SSD and liquid assisted extrusion for NC containing polyamide based systems. The approach is really interesting and deserves to be promptly distributed to the attention of the reader, being of extreme importance for processing of moisture sensitive and high melting polymers. According to this, I do not find any critical experimental result and/or discussion point, all the characterizations have been properly conducted and deeply explained, supported by appropriate literature.

I have only a doubt on the compression moulding step: could the authors better explain the reason why the "film was cut  into four pieces and placed on top of each other. Pre-heating for 60 s (at the same previous T?) was followed by pressing for another 60 s at the same pressure and temperature". I have understood the reason for the double heating to homogenize the PEG distribution, but I frankly do not understand  the 4 pieces layering.

Author Response

Reviewer 1

The authors presented the results of novel combination of SSD and liquid assisted extrusion for NC containing polyamide based systems. The approach is really interesting and deserves to be promptly distributed to the attention of the reader, being of extreme importance for processing of moisture sensitive and high melting polymers. According to this, I do not find any critical experimental result and/or discussion point, all the characterizations have been properly conducted and deeply explained, supported by appropriate literature.

Response: Dear reviewer, we are grateful for your positive feedback, thanks so much. The  revision we have made is  written  with a red. Please find our answers below.

I have only a doubt on the compression moulding step: could the authors better explain the reason why the "film was cut  into four pieces and placed on top of each other. Pre-heating for 60 s (at the same previous T?) was followed by pressing for another 60 s at the same pressure and temperature". I have understood the reason for the double heating to homogenize the PEG distribution, but I frankly do not understand  the 4 pieces layering.

Response: Thank you for your comment, we have modified the text in the manuscript for better understanding of this  second compression molding step. The final molding was made to ensure better homogenization of the  films and especially of the PEG-phase in the films. Therefore, the large film, after the first molding was cut into 4 pieces which were heated and pressed again, this layering is needed for the flow of the melt. Hope this clarifies why it was made as we have reported.

Reviewer 2 Report

This paper described water-assisted processing of PA6-cellulose nanocomposites, the drawing of those composites, and their characterizations. The paper is well-written and of interest to the community. 

The major weakness of the paper is the limited number of formulations/conditions examined. Any additional nanocellulose concentrations, processing conditions, etc. would be useful.

p.7, line 268: it's not clear what the authors mean by "PEG, which acts as moisture". Do the authors mean that PEG acts similarly to water?

The control sample (PA6) was also processed with water, which has previously been shown to affect PA6, including its crystal type and structure. A discussion about the effect of water on PA6 would be useful, along with any data showing neat PA6 properties without water processing.

The temperature values are hard to read in Figure 1a.

Author Response

This paper described water-assisted processing of PA6-cellulose nanocomposites, the drawing of those composites, and their characterizations. The paper is well-written and of interest to the community. 

Dear reviewer, we are grateful for your constructive feedback and revised our manuscript according to your comments.  The  revision is shown with red color and we hope that it is satisfactory.

The major weakness of the paper is the limited number of formulations/conditions examined. Any additional nanocellulose concentrations, processing conditions, etc. would be useful.

Response: Dear reviewer, initially we tested several combinations both NC (0.5, 1, 3 wt%) and PEG (5, 10, 15), the main goal was to find drawable nanocomposite. The nanocomposite with lowest NC concentration (0.5 wt%) with 5 wt% PEG was drawable, while larger concentration of NC reduced the drawability of the films. We don’t think that addition or all initial tests can be added into the manuscript.

Regarding your request of additional processing conditions, we have difficult to understand what kind of processing conditions are missing and would be useful. The information about the compounding process  describes the used NC concentrations in the liquid feeding process, extruder setup, temperature profile, screw speed, material flow rates, final material compositions, compression molding of the films and their drawing setups.

p.7, line 268: it's not clear what the authors mean by "PEG, which acts as moisture". Do the authors mean that PEG acts similarly to water?

Response: Thanks so much for this comment, we have modified this to  “The addition of PEG / NC did not affect the tan δ peak position, in contrast to the addition of NC only. The use of PEG was expected to move the tan δ peak to a lower temperature due to the plasticizing effect of PEG [31], but the NC in this material counteracts the effect and therefore the  peak position is not affected compared to PA6.”

The control sample (PA6) was also processed with water, which has previously been shown to affect PA6, including its crystal type and structure. A discussion about the effect of water on PA6 would be useful, along with any data showing neat PA6 properties without water processing.

Response: It is possible that water affect the PA6 and this is the reason why the same processing conditions was used for the ref material.The water is necessary in the liquid-feeding of the NC and therefore it is not the main interest to compared without and with water. However, we have in earlier studies showed that water feeing in the extrusion process is not affecting PLA, its molecular weight even it is known that  PLA is  moisture sensitive.

The temperature values are hard to read in Figure 1a.

Response: Thanks for pointing this, we have added the temperature setup in the text and also increased the font size in the  Figure 1a.

Reviewer 3 Report

The paper entitled Influence of Dispersion and Orientation on Polyamide-6 Cellulose Nanocomposites Manufactured through Liquid-assisted Extrusion is well written and organized. However, it needs to be revised before accepting for publication. My points of concerns are as follow

1: In Table 1 the authors show sample coding. They have utilized one concentration for each sample. However, I wonder how the authors came to conclusion that the sample with 0.5 % NC and/ or 5 % PEG are the optimum one and if any other sample synthesized with a different Wt % will not be better then the one mentioned in Table1.

2: Fig 1 and 2 needs to be presented with better clarity.

3: Sharp the colors and increase line width in Fig.7

4: Increase the size of Figure 8 for better clarity

Author Response

Reviewer 2

The paper entitled Influence of Dispersion and Orientation on Polyamide-6 Cellulose Nanocomposites Manufactured through Liquid-assisted Extrusion is well written and organized. However, it needs to be revised before accepting for publication. My points of concerns are as follow

Response: Dear reviewer, we are grateful for your constructive feedback and have revised the manuscript based on your comments, the revision is made with red color. Please find our answers below.

1: In Table 1 the authors show sample coding. They have utilized one concentration for each sample. However, I wonder how the authors came to conclusion that the sample with 0.5 % NC and/ or 5 % PEG are the optimum one and if any other sample synthesized with a different Wt % will not be better than the one mentioned in Table1.

Response: Thank you  so much for this comment, your opinion is correct, the materials  presented were not the only material combinations we made. We tested different concentrations and the best ones were used for this orientation study. The nanocellulose (NC) concentration were  0.5, 1, and 3 wt%  and  PEG concentration were 0, 5, 10, and 15 wt%. The  materials for the orientation work were chosen as 0 and 5 wt% PEG and 0.5 wt% NC. The main reason for this was that drawing was not possible for higher concentration of nanocellulose.

2: Fig 1 and 2 needs to be presented with better clarity.

Response:  Thank you, we  have modified both the Fig 1 and Fig 2 for better clarity.

3: Sharp the colors and increase line width in Fig.7

Response: Thank you, we have modified the Fig 7.

4: Increase the size of Figure 8 for better clarity

Response: We have increased the size as much it was possible for the page setup.

Reviewer 4 Report

The paper displays a series of results about Polyamide-6 Cellulose Nanocomposites, but it is need some improvement.

  1. In article: (Line119-121) Optical microscopy (OM) (Eclipse LV100POL, Nikon, Kanagawa, Japan) was used to observe the morphology of the fibers (Figure S1d) by placing a drop of the NC suspension (0.01 wt.%) on a glass. But in supplementary (Line19-20) Figure S1. (d) SEM image of NC suspension (0.01 wt.%), 19 scale bar 500 µm.

Which one is right?

  1. Before SSD, is there γ-phase in PA6 and PA6-nanocomposites? But we can’t get this information from the 1D-XRD of PA6 and PA6-nanocomposites. Why Tm1 of them attributed to the γ-phase not the α2? Maybe the FTIR of PA6 and PA6-nanocomposites before and after SSD from 500-1300 cm-1 can give more information about the crystal types.
  2. What is the effect of CNCs? Do the CNCs is the nucleation agents for crystallization? If the CNCs are nucleation agents for crystallization, why the addition of NC increased the apparent crystallite size in PA6, from 5.32 to 7.79 nm, and the addition of PEG in the nanocomposites further increased the size to 6.70 nm (282-284). Generally, the nucleation agents will increase the number of crystallite and decrease the size of crystallite. If the CNCs are reinforcement, why the tensile strength of PA6/0.5NC is almost the same with the PA6 (Figure 7)?
  3. Why do you choose 0.5% of NC concentrations? The first experiment may be prepared PA6/NC nanocomposites with different NC concentrations and find the optimum concentration.
  4. In Figure S1b AFM micrograph of the nanocellulose (0.01 wt. %) and width distribution of the nanocellulose and 1d OM image of NC suspension (0.01 wt.%), scale bar 500 µm. Why NC aggregated? Is the sample not well prepared? The state of CNC can be characterized by the TEM and AFM.
  5. Why do you choose polyethylene glycol (PEG) in your work? How about the glycerol?

Please explain. Thanks

Author Response

Reviewer 3

The paper displays a series of results about Polyamide-6 Cellulose Nanocomposites, but it is needing some improvement.

Response: We are grateful for the feedback and comments and your great efforts to improve our manuscript, the revision is made with red color. Please find our answers below.

  1. In article: (Line119-121) Optical microscopy (OM) (Eclipse LV100POL, Nikon, Kanagawa, Japan) was used to observe the morphology of the fibers (Figure S1d) by placing a drop of the NC suspension (0.01 wt.%) on a glass. But in supplementary (Line19-20) Figure S1. (d) SEM image of NC suspension (0.01 wt.%), 19 scale bar 500 µm. Which one is right?

Response: Thank you so much for observing this mistake. The correct method is optical microscope (OM), this is corrected now.

  1. Before SSD, is there γ-phase in PA6 and PA6-nanocomposites? But we can’t get this information from the 1D-XRD of PA6 and PA6-nanocomposites. Why Tm1 of them attributed to the γ-phase not the α2? Maybe the FTIR of PA6 and PA6-nanocomposites before and after SSD from 500-1300 cm-1can give more information about the crystal types.

Response: Thank you so much for your relevant question, we have improved this part in the manuscript and updated the Fig S6 of FTIR, to 500-3500 cm-1 in the Supplementary information, where more information of different crystal types is visible. The PA6 contain both α and γ crystals before the SSD and some peaks are in same positions before and after SSD, indicating both crystals forms are present. The peak positions for these crystal types are referred to Rotter and Ishida 1992, https://doi.org/10.1002/polb.1992.090300508.

The Tm1 peak is related to γ crystals (as referred in our paper), it is found also in the DSC results, a weak shoulder is visible before the SSD but after the SSD the Tm1 shows as a stronger peak.

Regarding the XRD, the γ peak was not visible in XRD before SSD probably because the peak is weak and hidden by the α1 and α2 peaks, however the FTIR results showed presence of both before and after SSD and higher intensity of the γ was found after the SSD.

  1. What is the effect of CNCs? Do the CNCs is the nucleation agents for crystallization? If the CNCs are nucleation agents for crystallization, why the addition of NC increased the apparent crystallite size in PA6, from 5.32 to 7.79 nm, and the addition of PEG in the nanocomposites further increased the size to 6.70 nm (282-284). Generally, the nucleation agents will increase the number of crystallite and decrease the size of crystallite. If the CNCs are reinforcement, why the tensile strength of PA6/0.5NC is almost the same with the PA6 (Figure 7)?

Response:  Thank you for your comment about the crystallization. We are not sure if the reviewer  has recognized that the nanocellulose used in this study was nanofibers and not cellulose nanocrystals which are much smaller. The nanofibers used are at least 10x wider and  the length is in micrometer scale. We have added more information about  the crystallization in the manuscript as well as in the Supplementary information. We studied the crystallization in polarized optical microscope with a hot-stage. The films were melted at 232 °C and cooled to 200 °C which is crystallization temperature of the PA6. The crystallization stared after 30-40 seconds and completed in 90 seconds and we couldn’t see difference between PA6, PA6/0.5NC and PA6/5PEG/0.5NC. This crystallization study is added in the Supplementary information, Figure S5. We agree with the reviewer that usually the crystal size would be expected to be smaller if the additive is acting as nucleation agent. However, the results from this study showed that the crystallization rate of PA6 is very fast, and we cannot see that the addition of NC would increase the crystallization rate or the nucleation.

Regarding the comment on the reinforcement effect. We agree that the addition of the 0.5wt%  nanofibers is only increasing the modulus  before the orientation but after orientation the addition of  this low nanofiber content  together with PEG as dispersing aid resulted in better strength toughness and modulus.

  1. Why do you choose 0.5% of NC concentrations? The first experiment may be prepared PA6/NC nanocomposites with different NC concentrations and find the optimum concentration.

Response: Thank you  so much for this comment, the reviewer is right, these were not the only material combinations we made and studied. We tested different concentrations and the best ones were used for the orientation study. The nanocellulose (NC) concentrations were  0.5, 1, and 3 wt%  and the PEG concentrations were 0, 5, 10, and 15 wt%. The  materials for the orientation work were chosen as 0 and 5 wt% PEG and 0.5 wt% NC. The main reason for this was that drawing was not possible for higher NC concentration.

  1. In Figure S1b AFM micrograph of the nanocellulose (0.01 wt. %) and width distribution of the nanocellulose and 1d OM image of NC suspension (0.01 wt.%), scale bar 500 µm. Why NC aggregated? Is the sample not well prepared? The state of CNC can be characterized by the TEM and AFM.

Response:  Dear reviewer, this nanocellulose is industrially produced, it is not chemically separated nanofibrils  but mechanically ground. The material consists of long entangled fibrils with a width around 30 nm and these can’t be compared with cellulose nanocrystals or chemically isolated nanocelluloses which are much smaller (5 nm). When analyzed with AFM, we have measured  the width of the fibrils and in the AFM image in the Supplementary information, we can see some  bundles which are larger than the small nanofibers. This is expected because the fibrils will form bundles or aggregates in the drying process. It is possible to see that the most measured fibrils are around 30 nm which is a typical width for mechanically separated nanocellulose fibrils. Then the optical microscopy image shows the reality, there are some larger bundles or aggregate, and  here it is good to remember that we can’t see the nanofibers in this image because these are below the resolution limit of the OM. This image is presented to show the reality as we are not removing these agglomerated NC as many other are doing, we can expect that same agglomerates are also present  in the nanocomposites. We are using  this NC because we are interested to work on larger scale and use materials which can be made industrially.

  1. Why do you choose polyethylene glycol (PEG) in your work? How about the glycerol? Please explain. Thanks

Response:  Thanks for your suggestion, the PEG was chosen as processing and dispersing aid because it is water soluble low molecular weight polymer and we have also in earlier work showed that use of PEG has  improved dispersion of the nanocellulose in PLA and enhanced the drawing of PLA. Regarding the glycerol, we have used glycerol triacetate in our previous studies with PLA and it is also working well as dispersing aid and plasticizer but is phase separating and an additional solvent was needed.

Round 2

Reviewer 3 Report

can be accepted

Reviewer 4 Report

The manuscript is improved and I have no questions. Thanks for the author’s response.